# Specific Urinary Metabolites in Malignant Melanoma

**DOI:** 10.3390/medicina55050145

**Published:** 2019-05-16

**Authors:** Marcela Valko-Rokytovská, Beáta Hubková, Anna Birková, Jana Mašlanková, Marek Stupák, Marianna Zábavníková, Beáta Čižmárová, Mária Mareková

**Affiliations:** 1Department of Chemistry, Biochemistry and Biophysics, University of Veterinary Medicine and Pharmacy in Košice, Komenského 73, 041 81 Košice, Slovakia; 2Department of Medical and Clinical Biochemistry, Pavol Jozef Šafárik University in Košice, Faculty of Medicine, Tr. SNP 1, 040 11 Košice, Slovakia; jana.maslankova@upjs.sk (J.M.); marek.stupak@upjs.sk (M.S.); beata.cizmarova@upjs.sk (B.Č.); maria.marekova@upjs.sk (M.M.); 3KOREKTCHIR s.r.o., Poliklinika Terasa, Toryská 1, 041 90 Košice, Slovakia; marianna.zabavnikova@gmail.com

**Keywords:** HPLC, urine, malignant melanoma, metabolite, 5,6-dihydroxyindole-2-carboxylic acid, vanilmandelic acid, homovanilic acid, tryptophan, 5-hydroxyindole-3-acetic acid, indoxyl sulphate

## Abstract

*Background and objectives:* Melanin, which has a confirmed role in melanoma cell behaviour, is formed in the process of melanogenesis and is synthesized from tryptophan, L-tyrosine and their metabolites. All these metabolites are easily detectable by chromatography in urine. *Materials and Methods:* Urine samples of 133 individuals (82 malignant melanoma patients and 51 healthy controls) were analysed by reversed-phase high-performance liquid chromatography (RP-HPLC). The diagnosis of malignant melanoma was confirmed histologically. *Results:* Chromatograms of melanoma patients showed increased levels of 5,6-dihydroxyindole-2-carboxylic acid, vanilmandelic acid, homovanilic acid, tryptophan, 5-hydroxyindole-3-acetic acid, and indoxyl sulphate compared to healthy controls. Concentration of indoxyl sulphate, homovanilic acid and tryptophan were significantly increased even in the low clinical stage 0 of the disease (indoxyl sulphate, homovanilic acid and tryptophan in patients with clinical stage 0 vs. controls expressed as medium/ interquartile range in µmol/mmol creatinine: 28.37/15.30 vs. 5.00/6.91; 47.97/33.08 vs. 7.33/21.25; and 16.38/15.98 vs. 3.46/6.22, respectively). *Conclusions:* HPLC detection of metabolites of L-tyrosine and tryptophan in the urine of melanoma patients may play a significant role in diagnostics as well as a therapeutic strategy of melanoma cancer.

## 1. Introduction

Cutaneous melanoma is a cancer that originates from the melanin-producing cells, the melanocytes of the skin’s epidermis. It is currently the deadliest type of skin cancer accounting for more than 75% of all deaths from skin cancer [1], the fifth and sixth most common solid malignancy diagnosed in men and women, respectively [2] and the nineteenth most common cancer type worldwide [1]. Its incidence rises with an increasing global rate per year at 2–7% annually [3].

Pigmented form of malignant melanoma is a tumour with an exceptionally high production of melanin. Hence, precursors of melanin and their metabolites play an important role in melanogenesis. In melanocytes, melanins are synthesized within melanosomes from L-tyrosine. It can be hydroxylated to L-dihydroxyphenylalanine (L-DOPA) and oxidized to dopaquinone (melanogenic pathway). Dopaquinone gives dopachrome in the following step through rapid cyclization. Dopachrome is then either decarboxylated to 5,6-dihydroxy-indole (DHI) or tautomerized to 5,6-dihydroxyindole-2-carboxylic acid (DHICA). DHI and DHICA are then oxidized to form eumelanin (dark brown to black pigment) [4,5].

L-tyrosine is, inter alia, a precursor of catecholamines (norepinephrine and epinephrine) and their degradation products: vanilmandelic and homovanilic acids (VMA, HVA). The biological role of catecholamines in melanogenesis is poorly understood, but in previously published articles it has been suggested that catecholamines could modulate mammalian melanogenesis [6,7]. According to the review of Sugumaran and Barek [8] enzymes as tyrosinase and the phenoloxidases involved in this conversion exhibit wide substrate specificity and oxidize a number of catecholic compounds. It is expected that a wide variety of endogenous catecholamines can serve as their substrates to participate in melanin biosynthesis.

Several lines of evidence demonstrate that activation of tumour b-adrenoreceptors can promote malignant progression by facilitating tumour survival, angiogenesis, migration, proliferation [9].

The efforts of earlier studies were focused on research on melanogenesis mechanisms. It has been shown that in addition to tyrosine, also tryptophan (Trp) and some of its metabolites are involved in the biosynthesis of melanins [10]. Serotonin and its metabolites occurring in urine, as 5-hydroxyindole-3-acetic acid (5-HIAA) are derived from tryptophan and act as specific tumour markers. Increased production of 5-HIAA has been documented in human epidermal keratinocytes and melanoma cells [11].

Melanoma is one of the fast-growing types of cancers worldwide. Prevention and early detection represent the only effective approach to reduce its incidence.

The aim of the present work was to develop simultaneous, qualitative and quantitative, high-performance liquid chromatography (HPLC) detection of metabolites of tyrosine (DHICA, VMA, HVA), Trp, its metabolites (5-HIAA and indoxyl sulphate (IS)) in urine of malignant melanoma patients as a tool of non-invasive melanoma diagnosis.

## 2. Materials and Methods

### 2.1. Composition of the Study Group

The study group consisted of 82 patients with pigmented malignant melanoma (clinical stage 0–IV; average age 57 ± 2 years; men and women ratio 44:38), and 51 healthy controls (average age 37 ± 11 years; men and women ratio 35:16) in a total volume of 133 people, with an average age of 49 ± 17 years, with a minimum of 17 years and a maximum of 86 years. Categorization of the volunteers was based on this selection with respect to melanoma clinical stages. Assigned numbers of the categories were as follows: 1–healthy control group; 2–malignant melanoma patients clinical stage 0; 3–malignant melanoma patients clinical stage IA and IB; 4–malignant melanoma patients clinical stage II, IIA and IIB; 5–malignant melanoma patients clinical stage III, IIIA and IIIB; 6–malignant melanoma patients clinical stage IV (Table 1).

Patients were recruited during hospitalization at the Department of Plastic and Reconstructive Surgery UPJŠ LF in Košice.

The diagnosis of malignant melanoma was confirmed histologically by the anatomic stage grouping for cutaneous melanoma using clinical staging (0–IV) on the basis of the Breslow thickness of the melanoma, the degree of spread to regional lymph nodes and by the presence of distant metastasis.

The healthy control group was chosen by random assignment, with the following criteria: absence of any disease, negative haematological and biochemical laboratory tests, negative family anamnesis.

Written informed consent was obtained from all patients prior to sample collection. All clinical investigations were conducted in accordance with the Declaration of Helsinki, and the study was approved by the Ethics Committee of the University of P.J. Šafárik in Košice, Medical Faculty (20N/2016).

### 2.2. Chemicals and Reagents

Acetonitrile, creatinine, VMA, HVA, Trp, 5-HIAA and indoxyl sulphate (IS) were purchased from Sigma–Aldrich (St. Louis, MO, USA), DHICA from Toronto Research Chemicals Inc. (Toronto, ON, Canada), formic acid from Riedel-de Haёn (Seelze, Germany). Ultrapure water was prepared by ultrafiltration of distilled water using the Simplicity system (Millipore, Molsheim, France). Mobile phase of 15% acetonitrile (ACN:H_2_O; 15:85) was prepared in deionised water with an addition of 0.05% formic acid. Stock solutions of creatinine, DHICA, VMA, HVA, Trp, 5-HIAA and IS were prepared by diluting given compounds to a concentration of 1 mg/ml in mobile phase. All reagents were of analytical grade.

### 2.3. Preparation of Standard Samples and Samples of the Experimental Study Group

Stock solutions of studied metabolites were diluted to obtain standard mixtures in the range of physiological and pathological values in human urine as follows: creatinine: 14–24–84–164 ppm; DHICA: 0.1–0.2–0.3 ppm; VMA: 0.5–1–5 ppm; HVA: 1–5–10 ppm; Trp: 1–5–10 ppm; 5-HIAA: 1–3–5; IS: 2–10–20 ppm.

Urine samples for HPLC analysis were obtained from patients with malignant melanoma immediately following admission to the hospital. Participants of healthy controls were given a morning appointment and asked to fast at least 8 h before the sample collection.

Urine samples were taken under standard conditions as first morning urine. Samples were stored at −27 °C. After thawing and centrifugation at 5000 rpm for 10 min at laboratory temperature (Centrifuge Boeco S8, Hamburg, Germany), samples were filtered by PVDF syringe filters with pore size of 0.2 μm and diluted with mobile phase. For HPLC analysis 5% urine was used.

### 2.4. Instrumentation

#### High-Performance Liquid Chromatography (HPLC)

HPLC was performed in modular reversed-phase high-performance liquid chromatography system (RP–HPLC, Schimadzu, Japan) with the use of Nucleosil expert column (EC) 100-5 C18 (Macherey–Nagel, Düren, Germany; column length 250 mm, inner diameter 4 mm, particle size 5 µm, pore size 100 Å). Samples were injected in a volume of 40 µL under isocratic conditions: mobile phase: 15% acetonitrile; flow rate: 0.8 mL/min; column temperature: 30 °C, analysis time: 30 min. Metabolites were detected by an ultraviolet–visible (UV–VIS) detector at 280 and 220 nm and by a fluorescence detector with a xenon lamp at excitation/emission wavelength 280/350 nm and 315/425 nm. The performance of the method was evaluated in terms of accuracy, linearity, imprecision and limit of detection.

### 2.5. Method Validation

Data were analysed using STATISTICA 10 data analysis software (StatSoft Inc., Tulsa, OK, USA).

Statistical analysis was performed via the following statistical tests: Pearson’s correlation test, Spearman rank order correlation test, Mann–Whitney U test of statistical significance. Pearson’s correlation test and Spearman rank order correlation test were used to determine the statistical dependence between parameters with normal and log-normal distribution, respectively. The ranges of urine metabolites were calculated according to their log-normal distribution.

### 2.6. Linearity of HPLC Method

The linearity of HPLC method was established by injecting standard mixtures of the metabolites in the range of physiological and pathological values in human urine in the concentration range 14–164 ppm for creatinine; 0.1–0.3 ppm for DHICA; 0.5–5 ppm for VMA; 1–10 ppm for HVA; 1–10 ppm for Trp; 1–5 for 5-HIAA and 2–20 ppm for IS. Representative regression equations for the calibration curves (*n* = 4) were y = 9.738 × 10^−6^x (R^2^ = 0.9999), y = 4.481 × 10^−8^x (R^2^ = 1.0000), y = 2.376 × 10^−6^x (R^2^ = 0.8740), y = 8.967 × 10^−6^x (R^2^ = 0.9755), y = 7.015 × 10^−8^x (R^2^ = 1.0000), y = 1.984 × 10^−7^x (R^2^ = 0.9993) and y = 5.384 × 10^−6^x (R^2^ = 0.9998) for creatinine, DHICA, VMA, HVA, Trp, 5-HIAA and IS, respectively.

### 2.7. Imprecision and Accuracy of HPLC Method

Inter-day method imprecision and accuracy were determined by replicate analysis (*n* = 2) of the same five urine samples in which the values of DHICA, VMA, HVA, Trp, 5-HIAA and IS were calculated on 10 consecutive days. The instrument imprecision was determined by repetitive injections of urine samples containing the same concentration of VMA (0.001 mg/L) and IS (0.01 mg/L) from the same vial performed on the same day.

### 2.8. Limit of Detection of HPLC Method

The limit of detection was determined by serial dilutions of working solutions to obtain a signal/noise ratio of approximately 3:1.

### 2.9. Clinical Application

The aim of the present work was to develop simultaneous qualitative and quantitative HPLC detection of DHICA, VMA, HVA, Trp, 5-HIAA and IS in urine of malignant melanoma and control group as a tool for non-invasive melanoma diagnostic.

## 3. Results

### Results of HPLC Analysis

DHICA, VMA, HVA, Trp, 5-HIAA and IS in mobile phase and in urine were analysed simultaneously by reversed-phase high-performance liquid chromatography (RP-HPLC) UV/VIS (280, 220 nm) and fluorescence detection (280/350 nm). For better resolution of DHICA, change in fluorescent detection to 315/425 nm was used. Chromatograms of melanoma and healthy urine samples are reported in Figure 1. Chromatogram of melanoma patient has increased peaks of all studied metabolites compared to chromatogram of healthy control. Peak integration and calibration were performed using LC Solution software (Shimadzu, Japan). Quantitative analysis was performed using a calibration curve obtained from injected standards to selected urine. The concentration of the monitored metabolites is expressed to measured creatinine level and is listed in Table 2.

Detections were performed by a fluorescence detector at the excitation/emission wavelength 280/350 nm for tryptophan, 5-hydroxyindole-3-acetic acid, indoxyl sulphate, homovanilic acid*, vanilmandelic acid* (A, C) and 315/425 nm for 5,6-dihydroxyindole-2-carboxylic acid (B, D). 1 = vanilmandelic acid*, retention time (t_R_) = 4.1 min; 2 = indoxyl sulphate, t_R_= 5.8 min; 3 = tryptophan, t_R_ = 6.9 min; 4 = 5-hydroxyindole-3-acetic acid, t_R_ = 7.5 min; 5 = homovanilic acid*, t_R_ = 9.5 min and 6 = 5,6-dihydroxyindole-2-carboxylic acid, t_R_ = 6.9 min (* and their metabolites).

The Pearson correlation test (for parameters with Gaussian distribution) and Spearman correlation test (for parameters with log-normal distribution, Table 3) were used to determine the relevance between urine metabolites and the factors for determining the prognosis for malignant melanoma, regardless of the clinical stage. The test pointed to a strong positive correlation between the urinary parameters and the presence of metastasis as well as the clinical stage of the disease, typically with high values of studied urine metabolites in malignant melanoma patients and low values in healthy control. However, the correlation between the prognostic factors Breslow thickness and Clark level was not significant. The Mann–Whitney U test confirmed statistical significance of the studied metabolites at respective stages of the disease when compared to healthy controls, but not at clinical stage 0 for DHICA, VMA and 5-HIAA. Significant differences with a *p* value below 0.001 were recorded between healthy control and the patients with a low clinical stage 0 for the parameters HVA (*p* = 2.29 × 10^−7^) and IS (*p* = 1.4 × 10^−13^).

Spearman’s cross-correlation between selected urine metabolites in µmol/mmol creatinine in healthy control and melanoma patients are listed in Table 4 and Table 5. There were no significant correlations detected between the studied parameters in healthy controls, except for Trp and its metabolite, IS (r = 0.371; *p* = 7.40 × 10^−3^, Table 4). By contrast, due to the increased concentration of the selected urinary metabolites in melanoma patients, almost every parameter correlated to each other: DHICA with VMA; VMA with HVA, Trp, 5-HIAA, IS; HVA with 5-HIAA and IS; Trp and 5-HIAA with IS (Table 5).

## 4. Discussion

In humans, the visible pigmentation of the skin that is in relation to its melanin content has a significant effect on the skin’s protective function against UV radiation damage. It is scientifically proven that people with lighter skin colour are exposed to a 40-fold greater risk of developing skin cancer than dark-skinned people.

Melanocytes are producing and excreting numerous metabolites, such as DHI and DHICA, with a specific role in the defence system against reactive oxygen species as well as in support of inflammatory and immune systems. Our results showed higher urinary excretion of DHICA (intermediate of eumelanin) in melanoma patients compared to control subjects (melanoma patients with clinical stages IA–IB and III–IIIA–IIIB, both *p* ≤ 0.01). This indicates that urinary DHICA has great clinical value in the progression of melanoma. Dopachrome tautomerase (DCT), the melanogenic enzyme is known for its overexpression in human melanoma cells that are resistant to both chemotherapy and radiation therapy [12,13]. According to Pak and Ben-David [14] the resistance is mediated through the accumulation of DHICA, which activates phosphor-extracellular signal-related kinase (ERK)/mitogen-activated protein kinase (MAPK) pathway. Its activation leads to processes that result in increased resistance to cancer treatment directed to cancer cell DNA damage.

Yamada et al. [15] determined the eumelanin precursor metabolites 5-S-cysteinyldopa and DHICA plus 6-hydroxy-5-methoxyindole-2-carboxylic acid as a marker for melanoma. Their results confirmed the presence of DHICA in the urine of patients with diagnosed melanoma where metastases were present in the concentration above 1 μmol/day. In the case of patients with metastasis-free melanoma, they demonstrated excretion of DHICA into urine but always less than the value mentioned above.

Earlier melanoma studies have shown that the urinary O-methyl derivatives of indole precursors of eumelanin such as 5-hydroxy-6-methoxyindole, 6-hydroxy-5-methoxyindole and the corresponding 5,6-dihydroxyindole-2-carboxylic acid methoxy- derivatives are present at higher levels in melanoma patients compared to healthy individuals, while 5-methoxyindole-2-carboxylic acid is an approved biochemical marker for malignant melanoma [16].

On the other hand, DOPA secreted from melanoma cells may be rapidly decarboxylated to dopamine, which in turn is metabolized to catecholamines and their metabolites, HVA and VMA.

Catecholamines, as typical stress hormones, can promote the aggressive potential of melanoma tumour cells through the interaction with speciﬁc receptors [17,18]. There is a hypothesis about the existence of paracrine interaction whereby keratinocytes secrete epinephrine in response to UV irradiation, which could stimulate neighbouring β-adrenergic receptors on melanocytes to increased melanin synthesis [19]. Yang et al. [20] in their study support the hypothesis that the stress hormone, norepinephrine, can stimulate the aggressive potential of melanoma tumour cells by inducing the release of proteins, including vascular endothelial growth factor (VEGF). We confirmed statistically significant concentration of catecholamine metabolite VMA (*p* ≤ 0.001) in the urine of patients with malignant melanoma even at stages IA-IB compared to the healthy control group.

The investigation for understanding the melanogenesis mechanism undertaken in previous research showed that not only tyrosine but also tryptophan and some of its metabolites are involved in the biosynthesis of melanins [10]. It has been reported that UVA exposure of skin can be associated with increased serum serotonin and decreased melatonin levels already after a single radiation exposure [21].

In patients with malignant melanoma, urine tryptophan (median of 14.92–17.88 µmol/mmol creatinine) and 5-HIAA concentrations (median of 2.36–6.93 µmol/mmol creatinine) were found to be significantly higher than in healthy controls (3.46 µmol of tryptophan/mmol creatinine and 2.62 µmol of 5-HIAA/mmol creatinine). Statistical significance when correlating the presence of metastasis with both parameters was evaluated as *p* < 0.001 (1.42 × 10^−19^ for tryptophan and 1.17 × 10^−4^ for 5-HIAA). These results are consistent with the result of Weinlich et al. [22], who described decreased serum tryptophan concentration as a poor prognostic marker for a malignant melanoma patient.

On the other hand, increased levels of 5-HIAA in human epidermal keratinocytes and melanoma cells are declared by Slominski [11]. Seemingly, urinary concentration of the tryptophan metabolite 5-HIAA is currently used to monitor disease progression or response to treatment not only in patients with carcinoid disease [23,24], and breast cancer [25], but also in patients with melanoma.

Metabolism of tryptophan is carried out inter alia by intestinal microflora, which converts it to indoxyl sulphate (IS) in the liver. Indoxyl sulphate, a uremic toxin, is normally excreted into urine and is accumulated in the serum of chronic kidney disease patients and bladder cancer patients [26].

Our data show significantly increased concentration of IS in urine (*p* ≤ 0.001) of melanoma patients (median 28.37–35.44 µmol/mmol creatinine) compared with the healthy control group (median 5.00 µmol/mmol creatinine). The highest correlation coefficient in melanoma patients (Spearman rank order correlation test) has been found between HVA and 5-HIAA (Spearman’s rho = 0.513; *p* = 3.40 × 10^−6^) and VMA and HVA (Spearman’s rho = 0.500; *p* = 6.66 × 10^−6^). The high correlation rate of the studied urinary metabolites confirms their action in melanoma detection. Surprisingly, the highest correlation coefficient was detected between IS and malignant melanoma with metastasis present (*p* = 1.10 × 10^−23^).

Cross-correlation between the metabolites of the total study group has been evaluated for future purposes to narrow the selection of metabolites associated with malignant melanoma. The results confirm well-known interactions between VMA and HVA as between Trp and 5-HIAA resp. Trp and IS, due to their cross-linked metabolisms. What is interesting is the correlation between urinary excretions of DHICA, Trp and IS, all acting in melanogenesis, but without unknown links to their mutual metabolic pathways.

## 5. Conclusions

We can conclude that the use of HPLC in the detection of urinary metabolites associated with melanoma seems to be an appropriate method due to its sensitivity and non-invasive character. Our results suggest that the following urinary metabolites can significantly contribute to the determination of melanoma: 5,6-dihydroxyindole-2-carboxylic acid, vanilmandelic acid, homovanilic acid, tryptophan, 5-hydroxyindole-3-acetic acid and indoxyl sulphate. A statistically significant difference in the concentration of IS, HVA and Trp between healthy controls and patients with a low clinical stage 0 indicates the importance of their determination in melanoma diagnosis at the early stages of the disease. An interesting correlation with a correlation coefficient over 0.50 appears between catecholamine metabolites VMA/HVA and 5-HIAA/HVA (*p* ≤ 0.001). Cross-correlation between the metabolites of the total study group has been evaluated for future purposes to narrow the selection of metabolites associated with malignant melanoma.

## Figures and Tables

**Figure 1 medicina-55-00145-f001:**
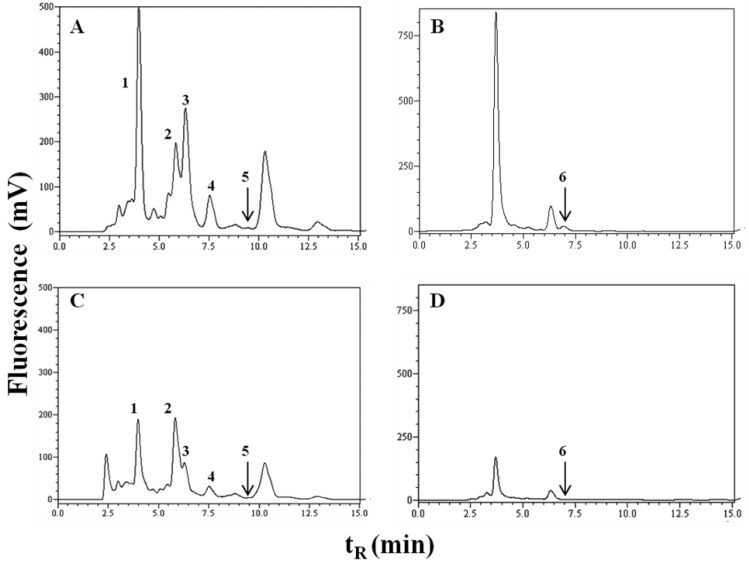
Chromatograms of urine: (**A**,**B**) melanoma patient; (**C**,**D**) healthy control.

**Table 1 medicina-55-00145-t001:** Descriptive statistics of the study group.

Clinical Stage	N	Age in Years		
		Minimum	Maximum	Mean
Healthy control	51	22	70	36.6 ± 10.9
Clinical stage 0	6	17	64	45.7 ± 18.1
Clinical stages IA–IB	28	18	79	54.9 ± 17.1
Clinical stages II–IIA–IIB	16	44	81	59.4 ± 11.2
Clinical stages III–IIIA–IIIB	23	30	86	59.9 ± 14.1
Clinical stage IV	9	34	78	58.1 ± 14.0

**Table 2 medicina-55-00145-t002:** Levels of selected urine metabolites assumed by reversed-phase high-performance liquid chromatography (RP-HPLC) expressed in µmol/mmol creatinine as median (and interquartile range).

Urine Metabolites	Healthy Control, *n* = 51	Clinical Stage				
		0, *n* = 6	IA–IB, *n* = 28	II–IIA–IIB, *n* = 16	III–IIIA–IIIB, *n* = 23	IV, *n* = 9
**DHICA**	0 (0.02)	0.12 (0.99)	1.42 (3.76)	2.19 (2.78)	1.72 (2.12)	2.56 (3.85)
		*p* = 0.096 ^ns^	*p* = 0.002 **	*p* = 0.041 *	*p* = 0.002 **	*p* = 0.037 *
**VMA**	13.87 (20.83)	32.82 (25.59)	37.67 (18.75)	41.29 (26.75)	38.74 (16.51)	39.86 (22.53)
		*p* = 0.073 ^ns^	*p* = 6.46 × 10^−6^ ***	*p* = 1.22 × 10^−4^ ***	*p* = 1.00 × 10^−6^ ***	*p* = 4.73 × 10^−6^ ***
**HVA**	7.33 (21.25)	47.97 (33.08)	81.41 (94.25)	71.77 (84.00)	82.94 (139.76)	68.20 (102.53)
		*p* = 2.29 × ^10−7^ ***	*p* = 6.98 × ^10−7^ ***	*p* = 0.004 **	*p* = 1.14 × ^10−4^ ***	*p* = 0.009 **
**Trp**	3.46 (6.22)	16.38 (15.98)	15.82 (7.89)	17.88 (11.90)	14.92 (8.93)	15.06 (9.82)
		*p* = 0.007 **	*p* = 1.1 × ^10−10^ ***	*p* = 1.63 × ^10−7^ ***	*p* = 1.16 × ^10−6^ ***	*p* = 0.005 **
**5-HIAA**	2.62 (2.02)	2.36 (4.18)	3.56 (3.72)	6.93 (4.51)	6.10 (2.10)	4.73 (4.22)
		*p* = 0.524 ^ns^	*p* = 0.042 *	*p* = 0.003 **	*p* = 4.08 × 10^−7^ ***	*p* = 6.30 × ^10−4^ ***
**IS**	5.00 (6.91)	28.37 (15.30)	33.34 (22.99)	35.44 (27.28)	29.35 (33.48)	30.81 (32.65)
		*p* = 1.4 × 10^−13^ ***	*p* = 6.5 × 10^−10^ ***	*p* = 6.49 × 10^−5^ ***	*p* = 1.21 × 10^−6^ ***	*p* = 7.65 × 10^−4^ ***

DHICA = 5,6-dihydroxyindole-2-carboxylic acid; VMA = vanilmandelic acid; HVA = homovanilic acid; Trp = tryptophan; 5-HIAA = 5-hydroxyindole-3-acetic acid; IS = indoxyl sulphate, *p* value of Mann–Whitney U test of urine metabolites in malignant melanoma patients versus healthy control, * correlation is significant at the 0.05 level; ** correlation is significant at the 0.01 level; *** correlation is significant at the 0.001 level; ns = correlation is not significant.

**Table 3 medicina-55-00145-t003:** Correlation test between the urinary parameters and the clinical parameters defining the respective clinical stages (Breslow thickness, Clark level, presence of metastasis).

Urine Metabolite		Breslow Thickness	Clark Level	Metastasis Present	Clinical Stage 0–IV
VMA (µmol/mmol of creatinine)	Pearson’s r	−0.011	−0.074	−0.268 **	0.177 *
	*p* value	0.926	0.539	0.002	0.042
5-HIAA (µmol/mmol of creatinine)	Pearson’s r	0.039	−0.014	−0.339 ***	0.298 ***
	*p* value	0.762	0.916	1.17 × 10^−4^	0.0008
DHICA (µmol/mmol of creatinine)	Spearman’s rho	0.064	0.187	−0.604 ***	0.610 ***
	*p* value	0.310	0.073	5.59 × 10^−14^	2.86 × 10^−14^
HVA (µmol/mmol of creatinine)	Spearman’s rho	0.038	−0.007	−0.729 ***	0.712 ***
	*p* value	0.383	0.477	4.13 × 10^−22^	9.95 × 10^−21^
Trp (µmol/mmol of creatinine)	Spearman’s rho	−0.014	0.020	−0.679 ***	0.666 ***
	*p* value	0.453	0.435	1.42 × 10^−19^	1.07 × 10^−18^
IS (µmol/mmol of creatinine)	Spearman’s rho	0.121	0.124	−0.730 ***	0.716 ***
	*p* value	0.155	0.152	1.10 × 10^−23^	1.89 × 10^−22^

Pearson’s r = Pearson’s correlation coefficient for parameters with Gaussian distribution (VMA and 5-HIAA) Spearman’s rho = Spearman’s correlation coefficient for parameters with log-normal distribution (DHICA, HVA, Trp and IS); VMA = vanilmandelic acid; 5-HIAA = 5-hydroxyindole-3-acetic acid; DHICA = 5,6-dihydroxyindole-2-carboxylic acid; HVA = homovanilic acid; Trp = tryptophan; IS = indoxyl sulphate; * correlation is significant at the 0.05 level; ** correlation is significant at the 0.01 level; *** correlation is significant at the 0.001 level.

**Table 4 medicina-55-00145-t004:** Spearman’s cross-correlation between selected urine metabolites in µmol/mmol creatinine in healthy control.

		VMA	HVA	Trp	5-HIAA	IS
**DHICA**	Spearman’s rho	0.033	0.108	−0.058	−0.155	0.168
	*p* value	0.817	0.450	0.685	0.277	0.239
**VMA**	Spearman’s rho		0.272	0.149	−0.034	−0.009
	*p* value		0.053	0.298	0.815	0.952
**HVA**	Spearman’s rho			0.004	0.111	0.192
	*p* value			0.976	0.439	0.177
**Trp**	Spearman’s rho				0.151	−0.371 **
	*p* value				0.291	7.40 × 10^−3^
**5-HIAA**	Spearman’s rho					−0.165
	*p* value					0.246

DHICA = 5,6-dihydroxyindole-2-carboxylic acid; VMA = vanilmandelic acid; HVA = homovanilic acid; Trp = tryptophan; 5-HIAA = 5-hydroxyindole-3-acetic acid; IS = indoxyl sulphate; ** correlation is significant at the 0.01 level.

**Table 5 medicina-55-00145-t005:** Spearman’s cross-correlation between selected urine metabolites in µmol/mmol creatinine in melanoma patients.

		VMA	HVA	Trp	5-HIAA	IS
**DHICA**	Spearman’s rho	0.234 *	0.196	0.216	−0.049	0.145
	*p* value	0.046	0.106	0.066	0.692	0.220
**VMA**	Spearman’s rho		0.500 ***	0.229 *	0.466 ***	0.326 **
	*p* value		6.66 × 10^−6^	0.039	3.22 × 10^−5^	0.003
**HVA**	Spearman’s rho			0.044	0.513 ***	0.302 **
	*p* value			0.711	3.40 × 10^−6^	0.009
**Trp**	Spearman’s rho				−0.074	0.245 *
	*p* value				0.536	0.026
**5-HIAA**	Spearman’s rho					0.386 **
	*p* value					7.41 × 10^−4^

DHICA = 5,6-dihydroxyindole-2-carboxylic acid; VMA = vanilmandelic acid; HVA = homovanilic acid; Trp = tryptophan; 5-HIAA = 5-hydroxyindole-3-acetic acid; IS = indoxyl sulphate; * correlation is significant at the Bonferroni adjusted 0.008 alpha level; ** correlation is significant at the Bonferroni adjusted 0.002 alpha level; *** correlation is significant at the Bonferroni adjusted 0.0002 alpha level.

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
