# Peer review of "Specific Urinary Metabolites in Malignant Melanoma"

_medicina, 2019, doi:10.3390/medicina55050145_

Round 1
Reviewer 1 Report
Dear Authors,
The article, "Specific urinary metabolites in malignant melanoma" describes an important observation, one that will have immediate clinical and therapeutic implications. The authors are congratulated on the good work. The study was designed with precision and several statistical analysis were done accurately.
The authors, however, need to sweep through the article to correct several places, where english language was out of context and the underlying meaning of the sentence was lost. There were also a few instances where space between two words in a sentence was missing, making the article appear a bit sloppy. The authors are also urged to revise the first sentence of their abstract. And finally, if the authors may change the abbreviation of table: Tab. to the full word, "table" in the text. This abbreviation is unnecessary and the full word will allow the sentences to flow better.
Author Response
Dear reviewer, thank you very much for your kind consideration, we really appreciate your comments and suggestions. Appropriate changes were made in the Abstract as well as in the text in relation to the tables.
Abstract - 1st sentence:
Original: Precursor of melanin, tryptophan, its metabolites as also the metabolites of L-tyrosine, which are easily chromatographically detectable in urine, are included in the process of melanogenesis with a confirmed modulator activity in this process.
Changed: Melanin, which has a confirmed role in melanoma cell behaviour, is formed in the process of melanogenesis and starts from tryptophan, L-tyrosine and their metabolites. All these metabolites are easily detectable by chromatography in urine.
„Tab“ – changed to „Table“
Reviewer 2 Report
Marcela Valko-Rokytovskáa et al. study the urinary metabolites in Melanoma patients using RP-HPLC. The manuscript was written well, need to revise well to address some issues before acceptance.
Major Comments:
1. what rationale to choose this study to examine specific amino acid urinary metabolite in melanoma patients?
2. Do you think this study need to link between tryptophan and neuro transmeters GABA?
3.I suggest strongly, if possible to include in vitro melanoma cell line baseline data in this study?
Minor Comments:
Abstract: The introductory lines need to be revised, to understand aim of this study in simple way?
Tables, figures need to revise, in publication quality?
Author Response
Dear reviewer, thank you very much for your kind consideration, we appreciate your comments and suggestions. Appropriate changes were made in the abstract.
Answer to Q1: Patients with melanoma are exposed to frequent blood taking. We were focusing on metabolite detection at the early stage of the disease that is possible to detect in urine, which is taken non-invasively.
Answer to Q2: Current knowledge suggests the importance of GABA in the proliferation and differentiation of several kinds of cells including cancer cells. It would be interesting to link these pathways, but in recent work, we focused on metabolites found in urine.
Answer to Q3: The research was performed using patient samples with melanoma and not cell lines.
Answer to minor comments: Appropriate changes were made in the abstract.
Reviewer 3 Report
The authors present interesting data about urine biomarkers for the detection of malignant melanoma. The putative biomarkers (metabolites of tryptophane, tyrosine or its metabolites) were identified and quantified by RP-HPLC. 82 melanoma samples were analyzed (different stages) and compared to 51 urine samples from healthy donors.
The following concerns should be addressed before publication.
Major points:
There exists no stage 0 in melanoma AJCC classification (8th edition) which should be used (compare tables 1, 2)
Multiple comparisons are made in the statistics. Therefore a correction like the Bonferroni method should be performed in order to correct the p values (alpha) for multiple testing (e.g. table 5). Maybe a statistician is needed to give some advice.
Since all biomarkers tested seems to be related to the melanin biosynthesis it would be interesting to know whethter the melanomas of the patients were all pigmented or included alemanotic melanomas. Is there an association with pigmentation of the melanomas or with the Fitzpatrick skin type (I-VI) of the patients?
Minor points:
Can the authors provide a single factor or a combination of factors that has a certain kind of sensitivity or specificity to detect melanomas (ROC analysis)?
The approval number of the Ethics Committee of the University of P. J. Šafárik in Košice should be provided in the Materials and Methods section.
Author Response
Dear reviewer, thank you very much for your kind consideration, we appreciate your comments and suggestions. Appropriate changes were made in the composition of the study group section and in the statistical analysis.
Answer to comment1: According to our knowledge, stage 0 is used to denote carcinoma/melanoma in situ with no metastatic potential. (https://www.ncbi.nlm.nih.gov/pmc/articles/PMC6230917/)
Answer to comment2: SPSS doesn’t offer Bonferroni adjusted p values for correlation analysis. However, we calculated the adjusted alpha and compared it with the SPSS offered p values for each correlation. Appropriate corrections were made in Table 5.
Answer to comment3: Amelanotic melanoma is a very rare form of cancer. Patient in our study were all patients with pigmented melanomas. We have fixed the patient specification in section 2.1. Composition of the study group.